Insights on the structure and stability of Licanantase: a trimeric acid-stable coiled-coil lipoprotein from Acidithiobacillus thiooxidans

Abarca Fernando 1 2
Gutierrez-Maldonado Sebastian E. 1 2
Parada Pilar 3
Martinez Patricio 3
Maass Alejandro 4 5
Perez-Acle Tomas 1 2 tomas@dlab.cl
1 Computational Biology Lab (DLab), Fundación Ciencia y Vida , Ñuñoa, Santiago , Chile
2 Centro Interdisciplinario de Neurociencias de Valparaíso (CINV), Universidad de Valparaíso , Valparaíso , Chile
3 Biosigma S.A. , Colina, Santiago , Chile
4 Mathomics, Center for Mathematical Modeling (CMM) and Center for Genome Regulation (CRG), Universidad de Chile , Santiago , Chile
5 Department of Mathematical Engineering, Universidad de Chile , Santiago , Chile
Uversky Vladimir
Electronic publication date: 2014 Aug 5
Publication date: 2014
Volume: 2
Electronic Location ID: e457
Received 2014 May 23; Accepted 2014 Jun 6
Copyright: © 2014 Abarca et al.
Copyright year: 2014
Copyright holder: Abarca et al.
License: This is an open access article distributed under the terms of the Creative Commons Attribution License, which permits unrestricted use, distribution, reproduction and adaptation in any medium and for any purpose provided that it is properly attributed. For attribution, the original author(s), title, publication source (PeerJ) and either DOI or URL of the article must be cited.
License URL: https://creativecommons.org/licenses/by/4.0/

Keywords: Bioleaching, Acidithiobacillus thiooxidans, Lipoprotein, Alanine-zipper, Protein structure prediction, Molecular dynamics simulation

Funding: PFB16 (FCV) FONDAP CRG 15090007 Millennium Institute Centro Interdisciplinario de Neurociencias de Valparaíso ICM-ECONOMIA P09-22-F CIRIC INRIA-Chile CONICYT This work was partially funded by PFB16 (FCV), FONDAP CRG 15090007, Millennium Institute Centro Interdisciplinario de Neurociencias de Valparaíso (ICM-ECONOMIA P09-22-F), CIRIC INRIA-Chile and supported by the supercomputing infrastructure of the NLHPC (ECM-02) “Powered@NLHPC”. Sebastian E. Gutierrez-Maldonado received a PhD scholarship from CONICYT. The funders had no role in study design, data collection and analysis, decision to publish, or preparation of the manuscript.

==============================
Licanantase (Lic) is the major component of the secretome of Acidithiobacillus thiooxidans when grown in elemental sulphur. When used as an additive, Lic improves copper recovery from bioleaching processes. However, this recovery enhancement is not fully understood. In this context, our aim is to predict the 3D structure of Lic, to shed light on its structure-function relationships. Bioinformatics analyses on the amino acid sequence of Lic showed a great similarity with Lpp, an Escherichia coli Lipoprotein that can form stable trimers in solution. Lic and Lpp share the secretion motif, intracellular processing and alpha helix structure, as well as the distribution of hydrophobic residues in heptads forming a hydrophobic core, typical of coiled-coil structures. Cross-linking experiments showed the presence of Lic trimers, supporting our predictions. Taking the in vitro and in silico evidence as a whole, we propose that the most probable structure for Lic is a trimeric coiled-coil. According to this prediction, a suitable model for Lic was produced using the de novo algorithm “Rosetta Fold-and-Dock”. To assess the structural stability of our model, Molecular Dynamics (MD) and Replica Exchange MD simulations were performed using the structure of Lpp and a 14-alanine Lpp mutant as controls, at both acidic and neutral pH. Our results suggest that Lic was the most stable structure among the studied proteins in both pH conditions. This increased stability can be explained by a higher number of both intermonomer hydrophobic contacts and hydrogen bonds, key elements for the stability of Lic’s secondary and tertiary structure.

Introduction

Bioleaching is a process with increasing interest for the mining industry. It consists of the release of heavy metals from insoluble ores through biological oxidation. Its importance relies on its low cost compared to flotation, smelting and conversion technologies that make the treatment of low-grade minerals feasible, thereby increasing the ore reserves available for exploitation. Moreover, bioleaching has less environmental impact than the traditional leaching process (Rawlings, 2004). Recently, Bobadilla Fazzini, Levican & Parada (2011) studied the secretome of gram-negative bioleaching bacteria Acidithiobacillus thiooxidans and Acidithiobacillus ferrooxidans, when grown in the presence of elemental sulphur. The major protein component of the secretome of Acidithiobacillus thiooxidans was a lipoprotein termed as Licanantase (Lic). Bioleaching assays showed that the addition of secretome fractions enriched in Lic resulted in an enhanced copper recovery from chalcopyrite (Bobadilla Fazzini, Levican & Parada, 2011). The authors suggested a possible participation of Lic as a surfactant agent, by removing the hydrophobic barrier formed by elemental sulphur over the surface of ores during chalcopyrite bioleaching. However, the mechanism by which the bioleaching rate is enhanced is not yet fully understood. Despite the authors reporting the amino acid sequence of Lic, the lack of structural information impeded further insights about its function. Within this context, our interest is to predict the structure of Lic in order to get insights on its structure-function relationships that may give us a glimpse on the role of this protein during bioleaching. To do so, we performed several bioinformatics analyses using the amino acid sequence of Lic. We found that, despite a low sequence identity, Lic shows a great similarity with a Lipoprotein (Lpp) of Escherichia coli and with its engineered mutants that contain larger alanine-zipper domains (Liu, Cao & Lu, 2002; Liu, Dai & Lu, 2003; Liu & Lu, 2002; Shu et al., 2000). Interestingly, Lpp forms trimeric coiled-coils in solution; nevertheless its poly-alanine mutants lose stability, as their alanine zipper domains are larger. Based on this evidence, we propose that the most probable structure for Lic is a trimeric coiled-coil. To further explore our structural hypothesis, we used de novo modelling to predict the 3D structure of Lic and Replica Exchange Molecular Dynamics (REMD) to assess its pH stability, using as controls the 3D structures of Lpp and a 14-alanine Lpp mutant (Ala-14). As a whole, our findings suggest that Lic exhibits sequence and structural features that improve its stability at the extreme low-pH environment where it performs its function. These features could be relevant to produce engineered versions of this protein in order to enhance the recovery of copper by bioleaching processes based on the presence of Acidithiobacillus thiooxidans.

Methods

Sequence-based analyses

Multiple sequence alignment was performed using ClustalW2 (Goujon et al., 2010; Larkin et al., 2007; McWilliam et al., 2013). The presence of known domains, destination signals and other sequence patterns was studied using ScanProsite, and LipoP and Signal Blast for destination signal (de Castro et al., 2006; Frank & Sippl, 2008; Rahman et al., 2008). TMHMM, TMPRED and TOPPRED were used to detect transmembrane domains and hydrophobicity profiles (Claros & von Heijne, 1994; Hofmann & Stoffel, 1993; Sonnhammer, von Heijne & Krogh, 1998; von Heijne, 1992). Secondary structure was predicted using Jpred, NPS@, PCI-SS, Porter and SCRATCH (Cole, Barber & Barton, 2008; Combet et al., 2000; Cheng et al., 2005; Green, Korenberg & Aboul-Magd, 2009; Pollastri & McLysaght, 2005). MultiCoil was used to predict coiled-coil regions (Wolf, Kim & Berger, 1997).

Cross-linking assay

E. coli cells that express recombinant Lic were treated with a hydrophobic or a hydrophilic cross-linking agent, DSP and DTSSP, respectively. PBS was used as control, which was the same solution used to solubilize the cross-linking agent. Proteins were analyzed by means of SDS-PAGE 15% and Western Blot His-probe. Samples were charged in non-denaturing conditions and denatured with β-mercaptoethanol.

Models

The Fold-and-Dock protocol, part of Rosetta 3.2 distribution, was used to predict the trimeric 3D structure of Lic using a cyclic symmetry (3C) (Das et al., 2009). Fragment libraries were generated by the Robetta server using the mature sequence of Lic composed of 80 amino acids residues (Kim, Chivian & Baker, 2004). 100,000 models were generated and clustered using a clustering application implemented in the Rosetta Suite. Briefly, this program selects the 400 lowest energy models and, by using a RMS cutoff of 3.0 Å, as suggested by Shortle, Simons & Baker (1998), the algorithm finds the structure with the largest number of neighbors within the cluster radius, creating a first cluster having this structure as the cluster center, including its neighbors in the cluster. The algorithm is repeated until all structures are assigned a cluster. The final model corresponds to the center structure of the highest populated cluster and it was named Lic-80.

Lpp-56 structure [PDB: 1EQ7] was completed using MODELLER (Eswar et al., 2007) by adding missing N-terminal cysteine and C-terminal lysine residues; the completed model was named Lpp-58.

The protonation states for titratable residues of Lic-80, Lpp-58 and Ala-14 [PDB: 1JCD] structures were assigned at acidic (pH 1.6) and neutral (pH 7.4) pH conditions using the PROPKA method (Li, Robertson & Jensen, 2005) as implemented in the PDB2PQR server (Dolinsky et al., 2007; Dolinsky et al., 2004), generating two models for each protein which were used as input for MD simulations. The difference between these models lies in the carboxyl groups being protonated at acidic pH, thus the modified amino acids are: D6, D15, E23, D37, D50, E64, E65, E68, E71 for Lic-80; D8, D13, D22, D27, D34, D40, D41, D50 for Lpp-58 and Ala-14; and C-terminal amino acids for all three proteins.

System setup for molecular dynamics and replica exchange simulations

In order to produce equilibrated 3D structures that could be used to conduct further structural analyses, molecular dynamics (MD) simulations were performed using the CHARMM27 force field in GROMACS 4.5.4 (Bjelkmar et al., 2010; Hess et al., 2008) for each of the protonated structures obtained in the previous step. To speedup the simulation process, the OBC implicit solvent approximation was used (Larsson & Lindahl, 2010; Onufriev, Bashford & Case, 2004). Each model was minimized using the Steepest Descent algorithm with a tolerance of 100 kJ mol−1 nm−1. Next, models were simulated for 500 ps using a harmonic restrain on alpha carbons with a constant force of 10 kJ mol−1 nm−2. Afterwards, each simulation was run without energetic restraints for 100 ns. The integration step was set to 2 fs. A stochastic dynamics integrator was used with a friction coefficient of 91 ps−1. Coulomb and Lennard Jones interactions were handled using a cutoff of 1.3 nm. Temperature was kept constant at 303.15 K using velocity rescaling with a stochastic term. Covalent bonds were constrained using P-Lincs algorithm (Hess, 2008). Trajectory frames were saved every 1 ps for analyses.

The final structures from previous simulations were used as the starting point for Replica Exchange Molecular Dynamics (REMD) simulations (Seibert et al., 2005; Sugita & Okamoto, 1999). To do so, 64 replicas for each system were simulated for temperatures ranging from 303.15 K to 809.57 K (303.15, 307.90, 312.72, 317.61, 322.59, 327.64, 332.77, 337.99, 343.29, 348.68, 354.15, 359.70, 365.35, 371.08, 376.92, 382.83, 388.86, 394.97, 401.18, 407.48, 413.91, 420.41, 427.03, 433.73, 440.55, 447.48, 454.52, 461.68, 468.95, 476.33, 483.83, 491.44, 499.18, 507.03, 515.01, 523.11, 531.35, 539.71, 547.96, 556.59, 565.34, 574.24, 583.28, 592.47, 601.81, 611.28, 620.84, 630.62, 640.54, 650.63, 660.87, 671.28, 681.85, 692.57, 703.47, 714.55, 725.79, 737.21, 748.80, 760.60, 772.55, 784.71, 797.04, 809.57). The temperature distribution was calculated using the “Temperature generator for REMD-simulations” server (Patriksson & van der Spoel, 2008), which was thoroughly tested by the authors who confirmed that predicted and observed exchange probabilities, one of the most important factors when performing REMD simulations, are correlated in 97%, with minor deviations due to the different force fields tested. Parameters were set for a desired exchange probability of 0.5. Each replica was simulated for 100 ns, obtaining a total combined time of 6.4 µs for each system. The exchange of replicas was attempted every 1 ps. Remaining parameters were the same ones used in the previous simulation step.

Simulation analysis

All the calculations were performed over coordinates extracted every 10 ps from trajectories.

RMSD and RMSF calculations of alpha-carbons were performed over entire MD trajectories. For RMSD, the first frame of each trajectory was used as reference. RMSF calculations were performed for each residue with final values corresponding to the average of the three monomers.

Hydrogen bonds, salt bridges, hydrophobic contacts and alpha helix content were measured over the last 50 ns of every MD trajectory and expressed as temporal averages with corresponding standard deviations.

Hydrogen bonds were calculated using cut-offs of 3.5 Å for donor–acceptor distance and 30° for the donor-H-acceptor angle. Salt bridges were calculated using a cut-off distance of 4.0 Å between donor–acceptor atoms (Barlow & Thornton, 1983). Unlike hydrogen bonds, salt bridges describe polar interactions that are independent of the geometry. Hydrophobic contacts were calculated using a cut-off distance of 7.0 Å between centers of mass for the side chains of hydrophobic residues. Only intermonomer interactions were considered for salt bridges and hydrophobic contacts, whereas for hydrogen bonds, intermonomer and main chain interactions were considered separately. Alpha helix content was calculated by means of the STRIDE program, as implemented in VMD (Frishman & Argos, 1995).

Unfolding temperatures (Tm) correspond to the minimum of the first derivative with respect to temperature for the observed variables measured during the last 50 ns of the REMD trajectories. This procedure was not possible when determining Tms for intermonomer H-bonds and salt bridges at neutral pH, thus mid points were calculated instead as data did not present a sigmoidal behavior (see Fig. S1). Calculations were performed using tools available in VMD and GROMACS.

Plots and statistical analysis were performed using R (R Development Core Team, 2011).

To reconstruct the conformational space at 303.14 K from the data collected at each temperature during the last 50 ns of REMD simulations, the weighted histogram analysis method (WHAM) was applied by using the Modular reweighting software (Sindhikara, 2011). Both the number of hydrophobic contacts and alpha helix content were used as the reaction coordinates for this analysis.

Results and Discussion

Sequence analyses

We searched for known domains and patterns using the amino acid sequence of Lic (Fig. 1). Our findings revealed the presence of a signal peptide located in the first 21 amino acids which could direct Lic to the membrane (Fig. 1, cyan box). Interestingly, a lipobox motif [L-(A/S)-(G/A)-C] (Hayashi & Wu, 1990) was detected with a potential cleavage site between residues 21|22 (Fig. 1, orange and blue boxes, respectively). Prediction of potential posttranslational modifications revealed that, after sorting to the inner membrane, the signal peptide could be cleaved and the free Cys22 could be anchored to the membrane by its attachment to a fatty acid (Hayashi & Wu, 1990; Tokuda, 2009). Previous reports suggest that for lipoproteins of gram-negative bacteria, the presence of an aspartate residue at position 2 (i.e., after cysteine 22) is a strong signal for the retention at the inner membrane, while any other amino acids promote its translocation to the outer membrane (Tokuda, 2009). Lic presents an alanine at position 2, thus Lic could be translocated to the outer membrane, which is in accordance to its presence in the secretome. No transmembrane domains were detected other than the region already predicted as a signal peptide, which is rich in hydrophobic residues (Fig. 1, red box). Secondary structure predictors coincided in the existence of two segments with an alpha-helix structure: the first one corresponding to the aforementioned signal peptide, whereas the second one corresponded to the processed mature protein (Fig. 1, pink cylinders).

Figure 1 Multiple alignment with related sequences and sequence-based predictions for Licanantase.

Multiple alignments are color-coded to show full (black background) or partial identity (grey background) between sequences. Letters on top of each alignment correspond to helical wheel diagram positions for each residue of Lic-80 (see Fig. 2 for details). Sequence-based predictions are characterized as follows: cyan box: signal peptide sequence residues 1–22; grey box: coiled-coil prediction; red box: transmembrane region residues 4–25; orange box: lipobox motif residues 19–22; blue box: cleavage site between residues 21|22; grey lines: coil secondary structure; purple cylinder: alpha helix secondary structure.

A FASTA (Pearson, 1990) search against the PDB database detected, as the best match, a 14-alanine mutant of Lpp [PDB: 1JCD] which shares a global sequence identity of 28.4% with Lic given mainly by the presence of an extended patch of alanine amino acid residues. Importantly, Lpp-56 [PDB: 1EQ7], which has an alanine-zipper domain composed of 3 alanines, and its mutants with extended alanine-zipper domains (i.e., Ala-5, Ala-7, Ala-10 and Ala-14 [PDB: 1KFM, 1KFN, 1JCC and 1JCD]), were shown to form trimeric coiled-coils when crystalized (Liu, Cao & Lu, 2002; Liu, Dai & Lu, 2003; Liu & Lu, 2002; Shu et al., 2000). According to Multicoil (Wolf, Kim & Berger, 1997) results, the sequence of Lic shows a probability of 0.99 to form coiled-coil structures and 0.78 to form trimeric coiled-coils in the second alpha helix segment. Notably, all predicted properties for Lic (i.e., processing, secondary structure and oligomerization state) coincided with those reported for Lpp (Shu et al., 2000). Considering the evidence as a whole, we propose that the most probable structure for Lic is a trimeric coiled-coil. A helical-wheel diagram for the proposed structure was plotted, where the hydrophobic residues can be seen aligned in positions a and d, forming a hydrophobic core with a large alanine-zipper domain (Fig. 2).

Figure 2 Helical wheel diagram of Licanantase.

Licanantase residues 22–101 can be arranged into 11 consecutive heptads where hydrophobic residues (mainly alanine) are located preferentially in positions a and d. Grey: hydrophobic residues. Orange: polar residues. Red: acidic residues. Blue: basic residues. Helical wheel was plotted using DrawCoil 1.0.

Oligomerization states

In order to further evaluate our structural hypothesis, cross-linking assays were performed on Lic, which was expressed heterologously in E. coli. These cells were treated with two cross-linking agents: DSP, a hydrophobic reagent that can cross cell membranes, and DTSSP, a hydrophilic reagent unable to cross membranes. DTSPP-treated cells showed the same band patterns as the DSP-treated ones (Fig. 3) suggesting that Lic can be found in the outer cell membrane which is in accordance to our bioinformatics analysis. Under native conditions, monomers, dimers and trimers can be observed, while on denaturing conditions the oligomeric forms are not present. The band patterns that were found (Fig. 3) resemble the patterns previously described by Choi et al. (1986) for Lpp when treated with cross-linking agents. These results provide further support for the predictions about the processing as well as the oligomeric state of Lic.

Figure 3 Cross-linking assay of Licanantase in E. coli.

Cells were treated with two cross-linking agents: DSP, a hydrophobic reagent that can cross cell membranes and DTSSP, a hydrophilic reagent unable to cross membranes. As a control, PBS solution was used. Proteins were analyzed in SDS-PAGE 15% (A) and Western Blot His-probe (B). Samples were charged in their native state (N) and denatured (D) with a reducing agent. The un-labeled lane in both images corresponds to the molecular weight standard. Black arrows in (B), from the top, indicate the position of the trimer, dimer and monomer, respectively.

Trimeric structure

Knowing that Lic can form trimers, the next question to answer was whether these trimers could form coiled-coil structures. To evaluate the possible trimeric conformations of the mature sequence of Lic, 100,000 models were generated using the Fold-and-Dock (Das et al., 2009) protocol of Rosetta. As described in the methods, the best 400 models were clustered, obtaining a main cluster with 71 members, in which all showed a coiled-coil structure. The center structure of this cluster was selected as the final model, and named as Lic-80 (Fig. 4). Root Mean-Square Fluctuation (RMSF) of alpha carbons was calculated in order to measure the structural variability among the members in the main cluster. The greatest variability among structures was found in their C- and N-terminal ends (Fig. 5). As previously proposed in Fig. 2, Lic-80 showed a hydrophobic core constituted mainly of alanines, forming an alanine-zipper domain. Lic-80 also has hydrophilic residues exposed to the solvent and located in positions where they can form electrostatic interactions between monomers (Fig. 4).

Figure 4 Licanantase model.

Trimeric Licanantase structure (Lic-80) was obtained by de novo prediction using its mature sequence of 80 aa. (A) Side view: from left to right, N-terminal to C-terminal end. (B) Top view: C-terminal end. Residue color coding: yellow, hydrophobic. Green, polar. Blue, basic. Red, acidic.

Figure 5 Root mean-square fluctuation of Licanantase models.

RMSF was calculated for alpha-carbon atoms over the 71 structures of the main cluster of Licanantase models.

Molecular dynamics simulations

To evaluate the structural stability of Lic-80 in forming a trimeric coiled-coil structure, MD simulations were performed at acidic and neutral pH. The 3D structures of Lpp-58 and Ala-14 were used as controls. Previous reports suggest that Lpp-58 loses structural stability in acidic pH conditions due to disruption of electrostatic interactions (Bjelić et al., 2008; Dragan et al., 2004). On the other hand, Ala-14 does not form stable structures in solution due to its mutations that extend the alanine-zipper domain to the entire hydrophobic core, being able to form trimeric coiled-coils only during the conditions applied for the crystallization process (Liu & Lu, 2002). It is important to note that, due to the aminoacidic composition of these proteins, the difference between MD simulations at acidic and neutral pH is given by the differential protonation states of the carboxyl groups, specifically for the side chains of aspartate and glutamate, and the free carboxyl group of the C-terminal amino acids.

As can be seen in Fig. 6, Root Mean-Square Deviation (RMSD) remained stable around 2 Å for both Lic-80 and Ala-14 and around 3 Å for Lpp-58, whereas only Lpp-58 showed a higher RMSD under acidic pH. RMSF profiles were similar for these three structures, showing higher fluctuations in their C and N-terminal ends, resembling the RMSF profile obtained for the main cluster of the Rosetta models (Fig. 5). In addition, Lpp-58 showed an increase in RMSF around Methionine 31 in acidic pH (Fig. 6). This phenomenon could be explained by the loss of intermonomer hydrogen bonds (H-bonds) between Arginine 32 and Aspartate 27 (Fig. 7), which would bring more flexibility to the trimers, thus allowing voluminous residues to be accommodated inside the hydrophobic core. This behavior could also explain the higher RMSD exhibited by Lpp-58 at acidic pH. On the contrary, the presence of an extended alanine-zipper domain in Ala-14 avoids the increase in RMSD and RMSF under acidic conditions by improving the packing of the coiled-coil. Our data shows this improved packing in terms of an increase in the number of hydrophobic contacts from 85 in Lpp-58 to 93 in Ala-14 at neutral pH and from 89 to 99 at acidic pH. Interestingly, the tightly packed coiled-coil has been previously reported by Liu & Lu (2002), where they observed a decrease in the supercoil radius (R0) for Lpp-56 from 6.1 Å to 5.1 Å in Ala-14.

Figure 6 Root mean-square deviation and root mean-square fluctuation of alpha carbon atoms during the entire simulations.

(A, D) Lic-80. (B, E) Lpp-58. (C, F) Ala-14. Red line: simulation at acidic pH. Blue line: simulation at neutral pH. Square dots and lines: average RMSF of the three monomers and its standard deviation.

Figure 7 Broken intermonomer H-bonds in Lpp-58.

Representative snapshots of Lpp-58 structure at acidic pH (A) and neutral pH (B). At acidic pH, Arginine 32 is unable to establish intermonomer interactions with Aspartate 27 because of its protonated state, while intramonomer interactions are occasionally seen with Asparagine 29. At neutral pH, both inter-/intramonomer interactions between Arginine 32 and Asparagine 29/Aspartate 27 can be seen. Red dotted lines: H-bonds formed between depicted residues.

In order to measure the effect of pH in nonbonding interactions, pH-sensitive and pH-insensitive interactions were measured. Those pH-sensitive interactions correspond to intermonomer H-bonds and salt bridges in which protonable carboxyl groups can participate, while pH-insensitive interactions correspond to main chain H-bonds and hydrophobic contacts. As expected, the number of intermonomer H-bonds and salt bridges found at neutral pH was higher than the ones found at acidic pH (Figs. 8A and 8B). However, the loss of electrostatic interactions due to acidic pH for Lic-80 (in average, 5 H-bonds and 6 salt bridges were broken) is less than that for Lpp-58 (in average, 18 H-bonds and 9 salt bridges were broken) and Ala-14 (in average, 21 H-bonds and 9 salt bridges were broken), thus the effect of an acidic environment over Lic stability should be lower.

Figure 8 Property averages for the last 50 ns of simulations of Lic-80, Lpp-58 and Ala-14.

(A) Intermonomer H-bonds. (B) Intermonomer salt bridges. (C) Intermonomer hydrophobic contacts. (D) Main chain H-bonds. (E) Helix content. Bars and lines: average and its standard deviation for each property at acidic pH (red) and neutral pH (blue).

Lic-80 showed a higher number of hydrophobic contacts than Lpp-58 and Ala-14, due to its longer extension producing a larger hydrophobic interface (Fig. 8C). Also, Lic showed the highest number of main chain H-bonds that stabilize the alpha helix structure. Interestingly, its alpha helix content did not change under different pH conditions (Figs. 8D and 8E).

During these simulations Ala-14 remained folded and the effect of acidic pH on Lpp-58 stability was less pronounced. These results proved the difficulty of leaving their potential energy wells and highlighted the need to use methods that can improve the exploration of conformational space.

Replica exchange molecular dynamics simulations

With the purpose of enhancing conformational space sampling, and being able to observe the unfolding process, we performed REMD simulations for a wide range of temperatures from 303.15 K to 809.57 K. To assess thermal stability, we analyzed the reduction of nonbonding interactions and alpha helix content in relation to the increase of temperature (see Fig. S1). Thermal stability was calculated in two different ways in order to better describe all the available data: the unfolding temperature (Tm) for data regarding hydrophobic contacts, main chain H-bonds and helix content at both pH conditions; and graph midpoints for intermonomer H-bonds and salt bridges at neutral pH. In terms of the obtained Tm and midpoint values (Table 1), the general unfolding process can be described as follows: first, intermonomer H-bonds and salt bridges are lost, followed by the loss of hydrophobic contacts, loss of the main chain H-bonds and, finally, overall disruption of secondary structure (Table 1, helix content). To graphically represent the unfolding process, we reconstructed the total conformational space explored in REMD simulations for each structure by applying the Weighted Histogram Analysis Method (WHAM) (Ferrenberg & Swendsen, 1989). For WHAM analysis, the number of hydrophobic contacts and alpha helix content, accounting for trimer and monomer stability respectively, were used as coordinates. The reconstructed conformational space was similar for the three structures, showing two high-probability basins (Fig. 9). The first and most populated basin receives contributions from folded structures at low temperatures (Fig. 9, upper-right corner in each graph), while the second basin receives contributions from fully-unfolded structures at high temperatures (Fig. 9, lower-left corner in each graph). Importantly, stable intermediate structures were not found suggesting that the unfolding process is a one step process. All the three analyzed structures showed elevated thermal stability during simulations, as high temperatures were needed to induce the unfolding process. In particular, Lpp-58 showed a Tm of 539 K at neutral pH, versus 338 K, which corresponds to the reported experimental value (Dragan et al., 2004). This could be explained due to the implicit solvent approximation that was used, which has been previously described as producing an overabundance of alpha helices in secondary structures (Roe et al., 2007).

Figure 9 Representation of conformational space at 303.15 K obtained by WHAM for each REMD simulation.

Colored contour plot indicating the probability of finding a structure with X% of helix content and Y% of hydrophobic contacts at acidic pH (A–C) and neutral pH (D–F) for Lic-80 (A, D), Lpp-58 (B, E) and Ala-14 (C, F).

Table 1 Unfolding temperatures in REMD simulations.

	pH 7.4	pH 1.6	
	Midpoint (K)	Tm (K)	Tm (K)	
	Intermonomer
H-bonds	Intermonomer
salt bridges	Hydrophobic
contacts	Main chain
H-bonds	Helix
content	Hydrophobic
contacts	Main chain
H-bonds	Helix
content	
Ala-14	483.83	483.83	499.18	507.03	515.01	523.11	523.11	539.71	
Lpp-58	468.95	491.44	507.03	523.11	539.71	531.35	547.95	556.59	
Lic-80	407.48	483.83	556.59	556.59	556.59	556.59	565.34	574.24	

Dragan et al. (2004) reported that upon lowering pH conditions from neutral (7.4) to acidic (3.0), Lpp-56 had its Tm decrease from 338 K to 316 K. However, during the performed simulations, this expected decrease of Tm due to acidic pH was not observed. As previously noted, this behavior could also be explained by the use of implicit solvent in our REMD simulations. Even though the GB/SA model for implicit solvent considers both screening effects and surface tension, it does not take into account the effect of hydrogen bonds formed between solvent and protein, nor the effect of ions (Roe et al., 2007). Thus, we cannot rule out that the observed differences in Tm for Lic-80 at acidic and neutral pH are accurate. However, the actual difference in Tm for Lic should be smaller than that of Lpp, because its structure shows less dependence on pH-sensitive hydrogen bonds and salt bridges (Figs. 8A and 8B). This evidence, combined with that of Lic-80 showing the highest Tm among the studied structures, allows us to propose the mechanisms by which Lic remains stable as a trimeric coiled-coil in the acidic environments of the bioleaching media.

Ala-14 exhibited the smallest Tm values in both pH conditions. However, its thermal stability was greater than expected, as Ala-14 does not form trimers and only shows 20% of alpha helix at 273 K (Liu & Lu, 2002). Lic-80’s structure shares with Ala-14 a large alanine-zipper domain that was reported as being destabilizing (Liu, Cao & Lu, 2002; Liu & Lu, 2002), however Lic-80 showed the highest Tm values. This could be explained by its greater number of pH-insensitive nonbonding interactions: ∼160 main chain H-bonds versus ∼100 in Ala-14 (Fig. 8C); and ∼160 hydrophobic contacts versus ∼90 in Ala-14 (Fig. 8D). Thus, Lic-80 could be compensating for the weakness of these interactions by establishing a greater number of them. Unlike Ala-14, Lic-80 has large hydrophobic residues that could contribute stronger hydrophobic interactions, specifically four methionines and one phenylalanine per monomer at the C-terminal end. Moreover, it is expected that these larger hydrophobic residues, especially methionines, could improve the capability of Lic to act as a surfactant agent for elemental sulphur. This function, together with the unique set of characteristics of the amino acid sequence of Lic would be key to the copper recovery process during bioleaching.

Conclusions

In this work we showed both in silico and experimental evidence to support the notion that the tridimensional structure of Lic is a trimeric coiled-coil. Although Lic exhibits a long alanine-zipper domain, which has been reported as a destabilizing factor, it presented the highest structural stability among the studied proteins. Thus, Lic showed a larger number of pH-insensitive nonbonding interactions that would stabilize its structure and provide resistance to acid environments. Further studies are required to evaluate the participation of the phenylalanine and the four methionines residues at the C-terminal end in Lic’s possible function as a surfactant agent. As a whole, our experimental/theoretical study has contributed to get insights on the biophysical properties that allow Lic to remain stable in extreme pH conditions.

Supplemental Information

Figure S1 Mean percentage of structural properties versus temperature

Mean percentage of structural properties during last 50 ns in each REMD simulation, Lic-80 (a, d), Lpp-58 (b, e) and Ala-14 (c, f). Color coding: intermonomer H-bonds (black); intermonomers salt bridges (red); main chain H-bonds (orange); hydrophobic contacts (green) and alpha helix (blue).

Click here for additional data file.

The authors would like to acknowledge Claudia Pareja for her technical work involving Fig. 1 and Walter Diaz for proofreading our manuscript.

Additional Information and Declarations

Competing Interests

Author Contributions

Dr. Pilar Parada and Dr. Patricio Martinez are members of Biosigma S.A. and they declare competing interests. Dr. Tomas Perez-Acle serves as an Academic Editor for PeerJ.

Fernando Abarca conceived and designed the experiments, performed the experiments, analyzed the data, contributed reagents/materials/analysis tools, wrote the paper, prepared figures and/or tables, reviewed drafts of the paper.

Sebastian E. Gutierrez-Maldonado performed the experiments, analyzed the data, wrote the paper, prepared figures and/or tables, reviewed drafts of the paper.

Pilar Parada conceived and designed the experiments, analyzed the data, contributed reagents/materials/analysis tools, reviewed drafts of the paper.

Patricio Martinez conceived and designed the experiments, performed the experiments, analyzed the data, contributed reagents/materials/analysis tools, reviewed drafts of the paper.

Alejandro Maass conceived and designed the experiments, analyzed the data, reviewed drafts of the paper.

Tomas Perez-Acle conceived and designed the experiments, analyzed the data, wrote the paper, prepared figures and/or tables, reviewed drafts of the paper.

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
