# Peer review of "Insights on the structure and stability of Licanantase: a trimeric acid-stable coiled-coil lipoprotein from Acidithiobacillus thiooxidans"

_PeerJ, doi:10.7717/peerj.457_

## Round 0.1 · accepted · Accept

As you can see, both reviewers agreed that your manuscript is well designed and recommended acceptance of original submission. This is a rather unique and unusual event since not too many manuscripts are accepted without revision. Congratulations!

Reviewer 1 ·

Basic reporting

NO comments

Experimental design

No comments

Validity of the findings

No comments

Additional comments

In this manuscript entitled “Insights on the structure and stability of Licanantase: a
trimeric acid-stable coiled-coil lipoprotein from Acidithiobacillus thiooxidans by Abarca et al , the authors describe prediction of the 3D structure of lipoprotein from Acidithiobacillus thiooxidans. The prediction is based on the sequence similarity with Lpp, an Escherichia coli lipoprotein, that can form stable trimers in solution. Furthermore, their conclusion is supported by cross-linking experiments. The structure is modeled and its molecular dynamics properties are explored.

In my opinion, this work was done at a good scientific level. The conclusions are supported by the results obtained. The predicted protein has important industrial application.

Minor suggestions: Circular dichroism study could provide additional support. It would be nice to have a more detail discussion about how their structural conclusion can help to identify function (functionally important residues) of this protein.

·

Basic reporting

I find this work on Licantase(Lic) by Abarca et al well designed and performed. The outcomes are also valuable for the understanding of the function of protein.
Beside the insights on Lic the understanding of the Lpp is broadened as well.

Experimental design

The experiments are designed and performed at the state of art level.
Only minor point considering attention are the expression and cross linking experiments. The blots should be performed in a more "professional" way (meaning lover voltage/current) or at least they should pass the software corrections for the rather high background.

Validity of the findings

The validity of the finding are given: positive and negative controls were applied as needed.
Minor point that should be addressed or be a topic of a future paper: the cross linking experiments give just a hint for a trimeric nature of the Lic, but no proof. Therefore a analytical ultracentrifugation experiment accompanied with circular dichroism one would give a solid proof for the theoretical findings.

Additional comments

The preparation of the representative structures for the unfolding pathway would provide explanations of the stability of those trimeric proteins. A compressed file containing those structures could be a part of Supplementary Information.